# Usefulness of Systemic Venous Ultrasound Protocols in the Prognosis of Heart Failure Patients: Results from a Prospective Multicentric Study

**DOI:** 10.3390/jcm12041281

**Published:** 2023-02-06

**Authors:** Marta Torres-Arrese, Arantzazu Mata-Martínez, Davide Luordo-Tedesco, Gonzalo García-Casasola, Rodrigo Alonso-González, Esther Montero-Hernández, Marta Cobo-Marcos, Beatriz Sánchez-Sauce, Valentín Cuervas-Mons, Yale Tung-Chen

**Affiliations:** 1Department of Emergency Medicine, Hospital Universitario Fundación de Alcorcón, Calle Budapest 1, 28922 Alcorcón, Spain; 2Department of Emergency Medicine, Hospital Universitario Infanta Cristina, Avenida 9 de Junio 2, 28981 Parla, Spain; 3Department of Radiology, Hospital Universitario Severo Ochoa, M-402, s/n, 28914 Leganés, Spain; 4Department of Internal Medicine, Hospital Universitario Puerta de Hierro–Majadahonda, Centro de Investigación en Red en Enfermedades Cardiovasculares (CIBERCV), Joaquín Rodrigo 1, 28222 Majadahonda, Spain; 5Department of Cardiology, Hospital Universitario Puerta de Hierro–Majadahonda, Centro de Investigación en Red en Enfermedades Cardiovasculares (CIBERCV), Joaquín Rodrigo 1, 28222 Majadahonda, Spain; 6Department of Internal Medicine, Hospital Universitario Fundación de Alcorcón, Calle Budapest 1, 28922 Alcorcón, Spain; 7Department of Internal Medicine, Hospital Universitario La Paz, Paseo Castellana 241, 28046 Madrid, Spain; 8Department of Medicine, Universidad Alfonso X, 28691 Madrid, Spain

**Keywords:** point-of-care ultrasound, Doppler ultrasound, VExUS, venous excess ultrasonography, acute heart failure, portal vein, hepatic vein, intra-renal vein, inferior vena cava

## Abstract

Mortality and re-admission rates for decompensated acute heart failure (AHF) is increasing overall and risk stratification might be challenging. We sought to evaluate the prognostic role of systemic venous ultrasonography in patients hospitalized for AHF. We prospectively recruited 74 AHF patients with a NT-proBNP level above 500 pg/mL. Then, multi-organ ultrasound assessments (lung, inferior vena cava (IVC), pulsed-wave Doppler (PW-Doppler) of hepatic, portal, intra-renal and femoral veins) were performed at admission, discharge, and follow-up (for 90 days). We also calculated the Venous Excess Ultrasound System (VExUS), a new score of systemic congestion based on IVC dilatation and pulsed-wave Doppler morphology of hepatic, portal and intra-renal veins. An intra-renal monophasic pattern (area under the curve (AUC) 0.923, sensitivity (Sn) 90%, specificity (Sp) 81%, positive predictive value (PPV) 43%, and negative predictive value (NPV) 98%), a portal pulsatility > 50% (AUC 0.749, Sn 80%, Sp 69%, PPV 30%, NPV 96%) and a VExUS score of 3 corresponding to severe congestion (AUC 0.885, Sn 80%, Sp 75%, PPV 33%, and NPV 96%) predicted death during hospitalization. An IVC above 2 cm (AUC 0.758, Sn 93.l% and Sp 58.3) and the presence of an intra-renal monophasic pattern (AUC 0. 834, sensitivity 0.917, specificity 67.4%) in the follow-up visit predicted AHF-related re-admission. Additional scans during hospitalization or the calculation of a VExUS score probably adds unnecessary complexity to the assessment of AHF patients. In conclusion, VExUS score does not contribute to the guidance of therapy or the prediction of complications, compared with the presence of an IVC greater than 2 cm, a venous monophasic intra-renal pattern or a pulsatility > 50% of the portal vein in AHF patients. Early and multidisciplinary follow-up visits remain necessary for the improvement of the prognosis of this highly prevalent disease.

## 1. Introduction

Acute heart failure (AHF) is a clinical syndrome whose overall incidence is increasing due to the aging of the population. The incidence rate in Europe is around 5/1000 person-years in the adult population, affecting 1–2% of the adults [1,2]. Mortality can be as high as 67% at five years after diagnosis [3]. Moreover, it is known that after diagnosis, patients with heart failure are hospitalized on average once a year [4]. Multidisciplinary programs had been implemented to tackle this high prevalence [5,6]. However, the optimal follow-up frequency is unknown. For this reason, tools are needed to refine patient prognosis.

The venous excess ultrasonography (VExUS) score is a new method of scoring systemic congestion based on inferior vena cava (IVC) dilatation and pulsed-wave Doppler (PW-Doppler) morphology of hepatic, portal and intra-renal veins. It has been proposed as a score to evaluate systemic congestion. The absence of congestion or VExUS 0 is considerate when IVC is smaller than 2 cm. Mild congestion or VExUS 1 is the presence of an IVC of almost 2 cm and any combination of normal (systolic wave greater than diastolic wave at hepatic vein PW-Doppler, pulsatility less than 30% at portal PW-Doppler, continuous pattern at intra-renal vein PW-Doppler) or mildly abnormal patterns (systolic wave smaller than diastolic wave at hepatic vein PW-Doppler, 30–50% pulsatility at portal PW-Doppler, biphasic flow at intra-renal vein PW-Doppler). The moderate congestion of VExUS 2 is considerate with a IVC of at least of 2 cm and one severely abnormal pattern at PW-Doppler morphology (S-wave reversal at hepatic vein PW-Doppler, >50% of pulsatility at portal PW-Doppler and discontinuous monophasic flow with only diastolic phase at intra-renal vein PW-Doppler). The presence of severe congestion (called VExUS 3) is due to an IVC of 2 cm or more and the presence of at least two severely abnormal PW-Doppler morphologies predicts acute kidney injury (AKI) in patients undergoing cardiac surgery. Its use has been adopted especially in heart failure patients, but to date has had no formal validation. Our study aims to evaluate different ultrasound parameters (including the VExUS score) in the prediction of clinically important outcomes (i.e., heart failure-related death, heart failure-related re-admission).

## 2. Materials and Methods

This is a prospective study performed in a tertiary and a secondary hospital. This study was conducted in accordance with the Helsinki Declaration and approved by our local Ethics Committee. We obtained informed consent from each patient.

### 2.1. Inclusion Criteria

Patients whose main hospital admission diagnosis was acute heart failure (AHF) and an NT-proBNP level above 500 pg/mL were included. Patients under 18 years of age, with hemodynamic instability, who declined to participate or who had received more than 24 h of diuretic treatment were excluded. Case inclusion was performed by four investigating physicians, different from the physician responsible for the patient. If an alternative pathology was diagnosed, the patient was excluded from the study. Patients who had an absence of echocardiographic findings consistent with heart failure decompensation, such as depressed left ventricular function or a high probability of diastolic dysfunction, were excluded from the study. Seventy-four patients who met the following inclusion criteria were prospectively studied (Figure 1).

### 2.2. Initial Assessment

Demographic data (age, sex, weight), comorbidities, risk factors for AHF (i.e., cardiopulmonary diseases), physical examination (weight, blood pressure, oxygen saturation), heart rate and laboratory tests (creatinine, urea, haemoglobin, leukocytes, NT-proBNP among others) were recorded at admission, discharge and follow-up visits. An early follow-up visit was scheduled in the outpatient clinic within the first fifteen days after discharge.

The EVEREST classification score (10), as a marker of the clinical course of congestion during hospitalization, was calculated for each patient at admission, discharge and consultation; the same was done with the NYHA dyspnea assessment scale. We defined worsening renal function as an increase of 0.3 mg/dL.

### 2.3. Collecting Ultrasound Data

Multi-organ ultrasound was performed in the first 24 h after admission, as well as on the day of discharge and in the follow-up within the first fifteen days after discharge (Figure 2). The different ultrasound parameters that could be associated with volume overload were registered:

Number of areas with pulmonary B-lines and pleural effusion (scored through the Lung Score, a lung involvement score that evaluates six areas per lung with a maximum score of 36) [7].

Diameter and collapsibility of the IVC. As in previous reported studies, we set the cut-off diameter at 2 cm [8].

PW-Doppler of hepatic, portal, renal and femoral veins [9].

Then, the VExUS score was obtained as described from previous reports [10]. In addition, based on a previous study from our research group [9], we calculated a simplified score from the previous categories (VExUS 0 = no congestion; VExUS 1 = mild congestion, VExUS 2 = moderate congestion; VExUS 3 = severe congestion) to two (0 = absence to moderate congestion, previous VExUS 0 to 2; 1 = severe congestion, previous VExUS 3).

Echocardiographic findings were also registered [9] (left ventricular diastolic and systolic diameters, interventricular septum and posterior wall), left ventricular ejection fraction, left and right atrial area, transmitral filling pattern, tricuspid annular plane systolic excursion (TAPSE), tricuspid regurgitation (TR) velocity, pulmonary artery diameter, right ventricular outflow acceleration time, pulmonary regurgitation velocity, presence of moderate or severe left valvular heart disease, and probability of pulmonary hypertension according to the European Society of Cardiology guidelines [11].

A Mindray M7 and M9 diagnostic ultrasound machine equipped with a phased, curvilinear, and linear transducer (Mindray España, Madrid, Spain) and a Kosmos portable ultrasound machine (EchoNous, Redmond, WA, USA) were used in the study.

### 2.4. Objective and Definitions

The main purpose of our study was to describe different ultrasound parameters and scores (including the popular VExUS systemic congestion score) in acute heart failure patients; and whether these parameters could predict complications, such as death and re-admissions related or not to heart failure. As a secondary endpoint, we assessed whether there are significant dynamic changes in these parameters during admission, and after treatment start.

We believe this pilot descriptive study, and the data obtained in this convenience sample, do not require a specific N.

## 3. Results

From November 2021 to August 2022, a total of 79 patients were evaluated and met the inclusion criteria (summarized in Figure 1 and Table 1), and 74 patients were included in the final analysis.

The mean age was 79.5 years (standard deviation—SD 12.5) and 51.4% were women. A total of 78.4% had an underlying cardiovascular disease and 43.2% had a previous pulmonary disease. Ten patients (13.5%) died during hospitalization, five (6.8%) within one month of discharge and four (5.4%) within two months. In total, 19 patients (25.7%) died, 12 deaths (16.2%) attributed to heart failure. Twenty-six patients (35.1%) were admitted in the first three months after discharge (15, 20.3% during the first month). Twenty-two patients (29.7%) had a heart failure-related re-admission (Table 1, second column). Regarding the echocardiographic findings on admission (Table 1, third column), probability of pulmonary hypertension was high in 49 patients (66.2%). Left ventricular function was severely reduced in 25 (33.8%) and TAPSE was reduced (TAPSE less than or equal to 17 mm) in 31 patients (41.9%).

Regarding clinical and laboratory findings (Table 2), most patients were in NYHA III on admission (62%) and improved to NYHA II upon discharge (37.8%), and NYHA I (37.8%) at follow-up visit to the clinic (Table 2). The median EVEREST score at admission was eight (SD 3.1), at discharge was 1 (SD 1.17) and at follow-up increased slightly to 2 (SD 2.43). The mean NT-proBNP was 10278.5 pg/L (SD 12740) on admission, 6156 pg/L (SD 7889) at discharge and 5438 pg/L (SD 5712) at follow-up. Mean creatinine was 1.58 mg/dL (SD 0.8) at admission, 1.47 mg/dL (0.94) at discharge and 1.52 at follow-up.

Among the dynamic changes on the ultrasound exams (Table 3), were remarkable the mean IVC diameter, which reduced from 2.25 cm (SD 0.53) to 1.81 cm (SD 0.42) from admission to discharge, and remained similar at early follow-up (1.85 cm, SD 0.53). The absence of collapsibility was present in 71 patients at admission (95.9%) and decreased at discharge (41 patients, 55.4%) and follow-up (38 patients, 51.4%). The dominant hepatic vein pattern at admission was systolic inversion, present in 35 patients (47.3%), improving in 26 patients (35.1%) during hospitalization. Portal vein pulsatility at admission was >50% in 27 patients (36.5%) and predominantly continuous (50 patients, 67.5%) at discharge. Regarding the intra-renal veins, the most observed pattern at admission was biphasic (30 patients, 40.5%) and was continuous at discharge (40 patients, 51.4%) and follow-up (31 patients, 41.9%). The dominant VExUS score on admission was 3 (24 patients, 32.4%). Remarkably at the same moment, 21 patients (28.4%) had a VExUS of 0, which increased to 56.8% at discharge, but decreased to 39.2% at follow-up. There were significant differences in all ultrasound parameters except for the collapsibility index between admission and discharge. However, this was not so significant between discharge and follow-up for the portal Doppler and VExUS score.

We collected significant (*p* < 0.01) moderate (r = 0.3–0.5) to strong (r > 0.5) correlations that are shown in Table 4. As we can see, the intra-renal Doppler assessment is likely to be the most useful marker, as it maintains the best correlation on admission, discharge, and follow-up.

Receiver operating characteristic (ROC) curve (Figure 3 and Appendix A) was calculated for predicting mortality (Figure 3a) and re-admission (Figure 3b) related to heart failure, based on the initial ultrasound exam.

The presence of an intra-renal monophasic pattern (area under the curve (AUC) 0.923, sensitivity (Sn) 90%, specificity (Sp) 81%, positive predictive value (PPV) 43%, negative predictive value (NPV) 98%) predicted death during admission, as did a VExUS score of 3 (AUC 0.885, sensitivity 80%, specificity 75%, PPV 33%, NPV 96%). A valid alternative could be the detection of a portal pulsatility > 50% (AUC 0.749, Sn 80%, Sp 69%, PPV 30%, NPV 96%).

To predict HF-related death, the VExUS of 3 had an AUC of 0.892 (Sn 92% and Sp 79%, PPV 46% and NPV 98%), similar to the simplified porta score (AUC 0.882, Sn 92%, Sp 74%, PPV 41%, NVP 98%) but less complex to calculate (Appendix A).

All ultrasound parameters were less accurate in predicting HF-related re-admissions. The best parameters were an IVC above 2 cm (AUC 0.758, Sn 93.l% and Sp 58.3) and the presence of an intra-renal discontinuous monophasic pattern (AUC 0. 834, sensitivity 0.917, specificity 67.4%) in the follow-up visit (Appendix A). However, it is interesting to find that the portal vein (AUC 0.696, *p* = 0.11) followed by VExUS (AUC 0.676, *p* = 0.23) at admission could also be of use (Figure 3b).

## 4. Discussion

Mortality and re-admission rates for decompensated acute heart failure (AHF) is generally increasing and risk stratification might be challenging, with a five-year mortality rate of over 60% [3] and one re-admission per year [4]. Given that it currently affects 1–2% of the population [1,2] and that its prevalence is increasing, to accurately stratify high-risk patients is of vital importance to the optimization of efforts and the allocation of resources.

Following our study results, and for prognostic classification purposes, repeating an ultrasound study during admission is most likely not necessary. We suggest performing an initial admission ultrasound, especially of the intra-renal veins. 

Previous studies have pointed out the role of ultrasound in accurately determining prognosis. Regarding the IVC, Cubo-Romano et al. reported that, in 80 patients hospitalized due to AHF, an IVC greater than 1.9 cm at admission had higher mortality rates at 90 days (25.4 vs. 3.4%; *p* = 0.009) and at 180 days (29.3 vs. 3.4%, *p* = 0.003) [8]. However, in the study of Beauvien-Souligny, the authors showed that the correlation with the IVC was poor [10], this could partly be explained due to the selected post-surgical population they recruited. Similar to our results, Goonwardena et al. followed a total of 75 patients admitted due to AHF and observed that the best predictors of re-admission were the IVC diameter of 2 cm (Sn of 81% and Sp of 72%) and the NT-proBNP (cut-off point of 2327 with a Sn of 82% and Sp of 56%) [12]. Moreover, Khandwalla et al. observed that each 0.5 cm increase in IVC diameter was associated with a 38% increase in the risk of re-admission (RR 1.38, *p* < 0.01) [13]. similar to our results, the AUC of the IVC at admission showed a trend towards significance in the prediction of re-admission and HF-associated death, probably due to the low sample size. We wish to highlight its usefulness in the follow-up visit after discharge, with a high NPV for mortality (Sn 100%, Sp 51%, PPV 38%, NPV 100%) and early re-admission (Sn 93%, Sp 60%, PPV 41%, NPV 97%).

Considering systemic congestion and its repercussion on the organs affected by it, Bouabda-llaoui et al. reported that portal pulsatility was associated with mortality in a cohort of 95 patients [14]. Similar to our results, measuring the portal vein at admission could be a good parameter for predicting death or early re-admission, as well as serve as a monitoring marker during follow-up. Therefore, our study emphasizes the potential role of the portal vein, with the advantage of its easy acquisition and reproducibility.

The data regarding intra-renal venous assessment are probably the most interesting. Husain-Syed prospectively evaluated 205 patients with RV failure undergoing cardiac catheterization and assessed congestion patterns and calculated the renal venous stasis index (RVSI), finding it to be prognostic [15]. Yoshihisa prospectively evaluated 314 patients and assessed both intra-renal arterial and venous components, observing that right atrial pressure was higher in monophasic than in non-monophasic, and that the cardiac event rate was higher in the low velocity time integral (VTI) and monophasic groups [16]. This is similar to our study, wherein we show an excellent AUC, and believe its high NPV could serve as a screening marker for identifying high-risk mortality and re-admission patients.

Beaubien-Souligny et al. [10] designed a venous congestion score (VExUS) in postoperative cardiac patients using the pulsed Doppler pattern of a hepatic vein, the portal vein and a renal interlobular vein [10], which had an association with the development of AKI. This system of congestion quantification has become widespread in the evaluation of AHF without data to support its use in guiding therapy or as a prognostic estimate.

To the best of our knowledge, this is the first study that assesses the role of VExUS score in the prognosis of AHF patients. For obvious reasons, we have focused on clinical outcomes rather than trying to define AKI in this population [17,18]. From our data we can state that the presence of a VExUS score of 3 on admission could predict death during admission, HF-related death, and early re-admission but is similar to other simpler ultrasound evaluations.

Another strength is that our study is one of the first reports to explore the frequency for which ultrasound should be performed in this population. We have analysed the ultrasound exam at admission, discharge, and follow-up, and found significant differences in the parameters from admission to discharge, but less differences between the discharge and follow-up. This would support the idea that serial ultrasound scans are not necessary and that the most cost-effective approach would be performing only an exam on admission and not repeating it during hospitalization.

On the admission ultrasound, portal and renal assessment would be an adequate predictor of mortality. Although the presence of a VExUS score of 3 may also be adequate, it does not contribute much more than the intra-renal or the portal veins, carrying a greater complexity and time-consuming in its evaluation. Consequently, if on the admission ultrasound we do not find severe intra-renal or portal patterns of congestion, we can probably classify this patient as a lower risk group of complications. During follow-up, the most reliable marker could be the size of the IVC, those below 2 cm being low risk.

It is important to acknowledge that our study has different limitations. First, in the tertiary hospital, there was an established heart failure program and in the secondary hospital, the follow-up was carried out by the internal medicine physician in charge, with a direct impact on the management. Second, patients living in long term care facilities with reduced mobility were less likely to have follow-up appointments, and therefore were less likely to be included. Thirdly, as this is a pilot study, only four expert sonographers were chosen to perform the exam, and, therefore, the results might not be reproducible. Future studies should investigate if the skill can be mastered by a greater number of novice sonographers. Finally, we have to highlight that the main purpose of the study was to compare the prognostic performance of different ultrasound parameters in the prediction of complications, and that the study was not designed to evaluate the performance of these parameters or the management strategy based on ultrasound. Therefore, for this purpose, the study can only be considered hypothesis generating and further studies on the prognostic implications of venous congestion ultrasound are needed to support the findings of this study. It would have been interesting to analyse the triggers of the AHF decompensation; however, after reviewing the patients’ electronic medical records, these were only found in 32.4% of cases, making it difficult to draw any conclusions.

Though these limitations are important, we believe that integrating ultrasound in our current practice is appropriate as it addresses more physiologically the assessment of the volume status in AHF patients.

## 5. Conclusions

The most cost-effective ultrasound scans are those on admission and at follow-up. Intra-renal venous Doppler assessment, VExUS score and the presence of a pulsatility above 50% on admission similarly predicts mortality. An inferior vena cava greater than 2 cm and an intra-renal monophasic pattern accurately predicts re-admission risk. VExUS probably adds unnecessary complexity to the assessment and prognosis of AHF patients. Early and multidisciplinary follow-up visits remain necessary to improve prognosis of this highly prevalent disease.

## Figures and Tables

**Figure 1 jcm-12-01281-f001:**
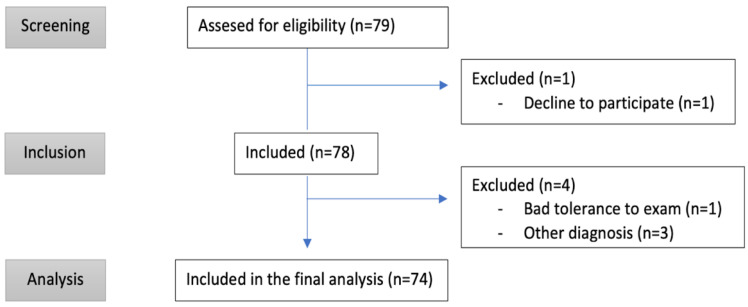
Strobe flow chart of the study.

**Figure 2 jcm-12-01281-f002:**
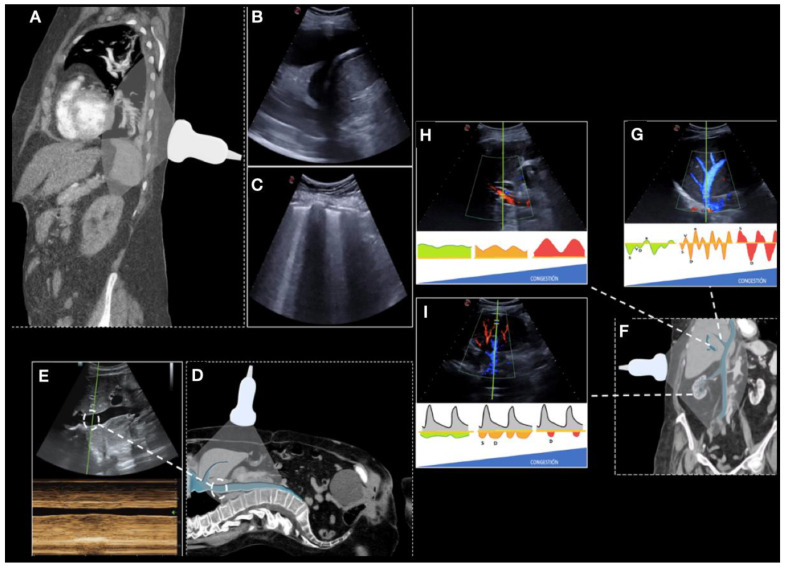
Description of the protocol study. Position of the probe in lung ultrasound (**A**) showing pleural effusion (**B**) and interstitial B-lines sign (**C**). Position of the probe in the inferior vena cava ultrasound (**D**) and an M-mode (**E**). Venous excess ultrasonography (VExUS, (**F**)), hepatic vein Doppler (**G**), portal vein Doppler (**H**), intra-renal vein Doppler (**I**).

**Figure 3 jcm-12-01281-f003:**
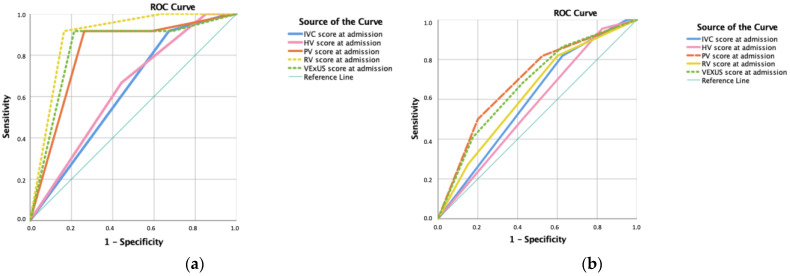
(**a**) Receiver operating characteristic (ROC) curve for predicting heart failure-related death according to intra-renal vein Doppler score at admission (area under the curve (AUC) 89.1%; *p* < 0.001) and VExUS score (AUC 83.2%; *p* < 0.001). (**b**) ROC curve for predicting heart failure-related re-admission according to portal vein score at admission (AUC 69.6%; *p* = 0.011) and VExUS score (AUC 67.6%; *p* = 0.023).

**Table 1 jcm-12-01281-t001:** Demographic characteristics, outcomes, and cardiac sonographic characteristics.

Demographic	N (%)	Outcomes	N (%)	Echocardiographic Findings	N (%)
Days of admission—mean (SD)	9.8 (4.72)	Death	19 (25.7)	TR	71 (95.9)
Age—mean (SD)	79.55 (12.5)	Death during hospitalization	10 (13.5)	Mild TR	18 (24.3)
Female (N, %)	38 (51.4)	HF-related death	12 (16.2)	Moderate TR	29 (39.2)
Male (N, %)	36 (48.6)	Death in the first month after discharge	5 (6.8)	Severe TR	24 (32.4)
Heart disease (N, %)	58 (78.4)	Death from day 30 to 90	4 (5.4)	Dilated RA	55 (74.3)
Atrial Fibrillation (N, %)	44 (59.5)	Unexpected visit	16 (21.6)	RV greater than LV	16 (21.6)
PH (N, %)	39 (52.7)	Admission in the first month	15 (20.3)	Anomalous movement of the sept	17 (23)
HFpEF (N, %)	56 (75.7)	Admission in the first three months	26 (35.1)	Outflow RV tract acceleration time < 105 ms	61 (82.4)
HFmrEF (N, %)	10 (13.5)	Admission related to HF	22 (29.7)	Pulmonary artery diameter > 25 mm	17 (23)
HFrEF (N, %)	8 (13.5)	Impaired kidney function	16 (21.8)	Low PH probability	14 (18.9)
Lung disease (N,%)	32 (43.2)	Admission hypertonic use	10 (13.5)	Intermediate PH probability	11 (14.9)
COPD (N, %)	14 (18.9)	Hypertonic use at follow-up	9 (12.2)	High PH probability	49 (66.2)
SAHS (N, %)	7 (9.5)	Vasoactive drugs	4 (5.4)	HFpEF	41 (55.4)
Interstitial disease (N, %)	0 (0)	Increase in diuretic treatment at follow-up	29 (39.2)	HFmrEF	8 (10.8)
Asthma (N, %)	9 (12.2)			HFrEF	25 (33.8)
Arterial hypertension (N, %)	65 (87.8)			TAPSE less than 17mm	31 (41.9)
Diabetes (N, %)	35 (45.9)				
Dyslipidemia (N, %)	33 (44.6)				
Advanced CKD (N, %)	27 (36.5)				
Obesity (N, %)	29 (39)				
Admission in the previous three months (N, %)	17 (23)				

Abbreviations: CKD = chronic kidney disease; COPD = chronic obstruction pulmonary disease; HF = heart failure; HFpEF = heart failure with preserved ejection fraction; HFmrEF = heart failure with mildly reduced ejection fraction; HFrEF = heart failure with reduced ejection fraction; LV = left ventricle; PH = pulmonary hypertension; RA = right auricle; RV = right ventricle; SD = standard deviation; SAHS = sleep apnea and hypopnea syndrome; TAPSE = tricuspid annular plane systolic excursion; TR = tricuspid regurgitation.

**Table 2 jcm-12-01281-t002:** Clinical and laboratory characteristics.

	On Admission	At Discharge	At Follow-Up
Systolic blood pressure—mean (SD)	136.31 (21.14)	121.27(17.80)	129.25 (25.08)
Diastolic blood pressure—mean (SD)	77.26 (17.79)	69.53 (12.18)	65.53 (11.25)
Oxygen saturation—mean (SD)	90.07 (7.46)	95.26 (2.31)	94.72 (4.21)
Weight (kg)—mean (SD)	72 (17.05)	69.45 (17.26)	70.86 (17.29
NYHA I (N, %)	0 (0)	37 (50)	28 (37.8)
NYHA II (N, %)	5 (6.8)	28 (37.8)	22 (29.7)
NYHA III (N, %)	46 (62.2)	1 (1.4)	8 (10.8)
NYHA IV (N, %)	22 (29.7)	0 (0)	2 (2.7)
EVEREST score—mean (DE)	7.76, 8 (3.1)	1.42 (1.17)	2.27 (2.43)
Urea—mean (SD)	74.99 (44.04)	86.98 (47.68)	72.76 (36.98)
Creatinine—mean (SD)	1.58 (1.03)	1.47 (0.94)	1.52 (0.83)
Creatine deteriorated—N (%)	55 (74.3)	59 (79.7)	59 (79.7)
Sodium—mean (SD)	138 (5.6)	139 (3.4)	138 (3.9)
NT-proBNP—mean (SD)	10278.5 (12740.7)	6156.14(7889.63)	5438.93 (5712.1)
GPT—mean (SD)	49.68 (48.17)	35.78 (25.18)	36.98 (14.2)
GOT—mean (SD)	41.90 (27.4)	28.27 (13.73)	31.90 (14.29)
Leukocytes—mean (SD)	8812.45 (3325.4)	7610.4 (2155.44)	7221.30 (2705.7)
Haemoglobin—mean (SD)	12.6 (2.28)	12.77 (2.14)	13.15 (2.14)

Abbreviations: GOT: glutamic oxaloacetic transaminase; GPT: Glutamate-Pyruvate Transaminase; NT-proBNP: NT-proB-type natriuretic peptide; NYHA: New York Heart Association; SD: standard deviation.

**Table 3 jcm-12-01281-t003:** Venous excess ultrasonographic protocol characteristics during admission, discharge and follow-up.

	On Admission (N = 74)	At Discharge (N = 64)	*p*-Value	At Follow-Up (N = 55)	*p*-Value
IVC (cm)–mean (SD)	2.25 (0.53)	1.81 (0.42)	<0.001	1.85 (0.43)	<0.001
Absence of collapsibility–N (%)	71 (95.9)	41 (55.4)	0.023	38 (51.4)	0.321
Lung score–mean (SD)	17.74 (7.23)	6.9 (5.62)	0.015	8.4 (7.8)	<0.001
Hepatic vein (SD)	1.34 (0.69)	0.98 (0.839)	<0.001	0.85 (0.81)	<0.001
S > D at hepatic vein flow–N (%)	9 (12.2)	20 (27)		23 (31.1)	
S < D at hepatic vein flow–N (%)	29 (39.2)	26 (35.1)		18 (24.3)	
S Reversal at hepatic vein–N (%)	35 (47.3)	19 (25.7)		14 (18.9)	
Portal vein (SD)	0.94 (0.839)	0.32 (0.612)	<0.001	0.47 (0.56)	0.19
Pulsatility < 30%–N (%)	27 (36.5)	50 (67.6)		33 (44.6)	
Pulsatility 30–50%–N (%)	20 (27)	11 (14.9)		23 (31.1)	
Pulsatility > 50%–N (%)	27 (36.5)	5 (6.8)		2 (2.7)	
Intra-renal vein (SD)	0.88 (0.734)	0.58 (0.74)	<0.001	0.64 (0.76)	<0.001
Continuous–N (%)	23 (31.1)	38 (51.4)		31 (41.9)	
Discontinuous Biphasic N (%)	30 (40.5)	18 (24.3)		17 (23)	
Discontinuous monophasic–N (%)	21 (28.4)	10 (13.5)		10 (13.5)	
VExUS score (SD)	1.50 (1.18)	0.65 (1.015)	<0.001	0.95 (1.09)	0.052
VExUS 0–N (%)	21 (28.4)	42 (56.8)		29 (39.2)	
VExUS 1–N (%)	13 (17.6)	12 (16.2)		8 (10.8)	
VExUS 2–N (%)	16 (21.6)	5 (6.8)		14 (18.9)	
VExUS 3 -N (%)	24 (32.4)	7 (9.5)		6 (8.1)	

Abbreviations: IVC: inferior vena cava; SD: standard deviation; VExUS: venous excess ultrasonography score.

**Table 4 jcm-12-01281-t004:** Statistically significant (*p* < 0.001) moderate-to-strong correlations of ultrasound findings.

Death	Death during Admission	HF-Related Death	Re-Admission	Re-Admission (First Month)	HF-Related Re-Admission
	IVC_a_ (r = 0.432)	IVC_a_ (r = 0.516)			
SPS_a_ (r = 0.320)	SPS_a_ (r = 0.357)	SPS_a_ (r = 0.504)		SPS_a_ (r = 0.363)	SPS_a_ (r = 0.317)
SI-rS_a_ (r = 0.440)	SI-rS_a_ (r = 0.540)	SI-rS_a_ (r = 0.618)		SI-rS_a_ (r = 0.393)	
VExUS 3_a_ (r = 0.377)	VExUS 3_a_ (r = 0.402)	VExUS 3_a_ (r = 0.557)		VExUS 3_a_ (r = 0.444)	
		SI-rS_d_ (r = 0.358)		SI-rS_d_ (r = 0.346)	
IVC_f_ (r = 0.438)				IVC_f_ (r = 0.442)	
			SSS_f_ (r = 0.426)	SSS_f_ (r = 0.399)	SSS_f_ (r = 0.356)
				SI-rS_f_ (r = 0.524)	
				VExUS 3_f_ (r = 0.453)	

IVC_a_ = Inferior cava vein at admission. SPS_a =_ simplified portal score at admission (pulsatility of >50%). SI-rS_a_ = simplified intra-renal score at admission. VExUS 3_a_ = VExUS 3 at admission. SI-rS_d_ = simplified intra-renal score at discharge. SSS_f_ = simplified hepatic score at follow-up. SI-rS_f_ = simplified intra-renal score at follow-up. VExUS 3_f_ = VExUS 3 at follow-up.

## Data Availability

The authors confirm that the data supporting the findings of this study are available from the corresponding author, upon reasonable request.

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
