# Peer review of "Usefulness of Systemic Venous Ultrasound Protocols in the Prognosis of Heart Failure Patients: Results from a Prospective Multicentric Study"

_jcm, 2023, doi:10.3390/jcm12041281_

Round 1

Reviewer 1 Report

I had the pleasure to review the paper entitled

“Usefulness of systemic venous ultrasound protocols in the 2 prognosis of heart failure patients: results from a prospective 3 multicentric study”

This article investigated the role of a comprehensive imagistic evaluation in patients with decompensated heart failure and found no additional value for the patient’s prognosis with the exception of dilated inferior vena cava greater than 2 cm and the venous monophasic intrarenal pattern, while a pulsatility > 50% of the portal vein may also be of interest.  

Very interesting was the pattern distribution of HF, with only one third of patients having HFrEF. It would be interesting to see if this comprehensive evaluation would have play a role in the outcome of this particularly severe category of patients.

On the other hand, since the mean systolic blood pressure at admission was 136±31mm Hg, the decompensation causes and the underlying structural disease, especially in patients with HFpEF should have been described.

The paper is very well written and has a solid statistic data analysis, and it certainly adds value to pre-existing published data

Author Response

Reply:

  • We greatly appreciate the reviewer efforts for the positive and constructive comments. We fully agree with you that it would had been interesting to report the reason for the cardiac decompensation. One variable in our study was whether a trigger was found. This search was carried out through the clinical history. We only found it in 32.4%. Since the investigating physician was not the treating physician, there is a probable bias in this information, since it is plausible that the responsible physician found triggers (non-compliance, therapy optimization, salty diet... that were not reflected in the clinical judgement). For this reason, we decided to omit it from the text at the time, but following your comment, we have added as a limitation point at the end of the discussion.
    • It would had been interesting to analyze the triggers of the AHF decompensation, however after reviewing the patient electronic medical record, these were only found in 32.4% of them, making it difficult to draw any conclusions.”

Reviewer 2 Report

The authors present a study evaluating the potential usefulness of a systemic venous ultrasound evaluation in the prognosis of patients hospitalized by AHF (including patients with preserved, mild-reduced, and reduced left ventricular ejection fraction).

Although the study is interesting, I have several comments about the study design.

The objective of the study is to evaluate the prognostic role of systemic venous ultrasonography in patients with AHF. For that, the authors have designed a prospective study including 74 consecutive patients. However, I think that the Methods section is not enough clear. 

Is mandatory to establish a specific primary endpoint (and if authors consider other secondary endpoints). Based on the primary endpoint, the authors should have calculated the sample size to achieve the primary endpoint. And Results section should to respond the different endpoints.

Inclusion criteria: ¿Was not mandatory to perform an echocardiogram previous to randomization to confirm the diagnosis of Heart Failure? The absence of structural cardiac abnormalities has a very high negative predictive value. I think the inclusion criteria should be clarified.

Discussion: Generally, the discussion begins with a paragraph that resumes the study's main findings to simplify to the lector the results of the study objectives.

In summary, in my opinion, the study has several major methodological errors that should be modified to improve the scientific quality.

Author Response

REVIEW REPORT 2.

The authors present a study evaluating the potential usefulness of a systemic venous ultrasound evaluation in the prognosis of patients hospitalized by AHF (including patients with preserved, mild-reduced, and reduced left ventricular ejection fraction).

Although the study is interesting, I have several comments about the study design.

The objective of the study is to evaluate the prognostic role of systemic venous ultrasonography in patients with AHF. For that, the authors have designed a prospective study including 74 consecutive patients. However, I think that the Methods section is not enough clear. 

Is mandatory to establish a specific primary endpoint (and if authors consider other secondary endpoints). Based on the primary endpoint, the authors should have calculated the sample size to achieve the primary endpoint. And Results section should to respond the different endpoints.

Reply:

  • We thank the reviewer for the comment. We agree that the aim was not clear. Therefore, we clarified and amended it, as followed:
    • “The main purpose of our study was to describe different ultrasound parameters and scores (including the popular VExUS systemic congestion score) in acute heart failure patients; and whether these parameters could predict complications, such as death and readmissions related or not to heart failure. As a secondary endpoint, we assessed whether there are significant dynamic changes in these parameters during admission, and after treatment start.”
    • As a pilot descriptive study, and the data obtained in this convenience sample, we believe it does not require specific N.”
    • Therefore, we added to the limitations section: “we have to highlight that the main purpose of the study was to compare the prognostic performance of different ultrasound parameters in the prediction of complications, and the study was not powered to evaluate the performance of these parameters or management strategy based on ultrasound, therefore, for this purpose, the study can only be considered hypothesis generating.”

Inclusion criteria: ¿Was not mandatory to perform an echocardiogram previous to randomization to confirm the diagnosis of Heart Failure? The absence of structural cardiac abnormalities has a very high negative predictive value. I think the inclusion criteria should be clarified.

Reply:

  • We appreciate your comment. In the material and methods inclusion criteria section, we stated:
    • “If an alternative pathology was diagnosed, the patient was excluded from the study”. In this sentence we meant that if, when performing the multiorgan ultrasound, we did not find a compatible ultrasound (that is, data of pulmonary congestion or right failure with an echocardiographic substrate – systolic dysfunction or high probability of diastolic dysfunction), the patient was excluded from the study. For this reason, all the included patients presented heart disease and systemic findings of congestion. To facilitate understanding we proceed to amend this sentence.

Discussion: Generally, the discussion begins with a paragraph that resumes the study's main findings to simplify to the lector the results of the study objectives.

Reply:

  • We thank again the reviewer for his/her helpful comment. I will start the Discussion section with a brief paragraph of introduction of the most relevant findings of our study, as suggested.

In summary, in my opinion, the study has several major methodological errors that should be modified to improve the scientific quality.

Reply:

  • We greatly appreciate the reviewer efforts for all the previous positive and constructive comments that we have incorporated to this new version. We hope that with the changes following the reviewer’s comments and suggestions, we can have clarified the questions arisen. We look forward to any other comments or questions.

Round 2

Reviewer 2 Report

Agree with your comments